# Short-Term Impact of Recycling-Derived Fertilizers on Their P Supply for Perennial Ryegrass (*Lolium perenne*)

**DOI:** 10.3390/plants12152762

**Published:** 2023-07-25

**Authors:** Lea Deinert, Israel Ikoyi, Bastian Egeter, Patrick Forrestal, Achim Schmalenberger

**Affiliations:** 1Department of Biological Sciences, School of Natural Sciences, University of Limerick, V94 T9PX Limerick, Ireland; lea.deinert27@gmail.com (L.D.); israel.ikoyi@ucd.ie (I.I.); 2School of Biology and Environmental Sciences, University College Dublin, D04 V1W8 Dublin, Ireland; 3CIBIO-InBIO, Campus de Vairão, Universidad e do Porto, 4485-661 Vairão, Portugal; bastian.egeter@naturemetrics.co.uk; 4Department of Environment, Soils and Land-Use, Teagasc, Johnstown Castle, Y35 Y521 Wexford, Ireland; patrick.forrestal@teagasc.ie

**Keywords:** *Lolium perenne*, nutrient cycling, phosphorus, *phoD*, recycling-derived fertilizers

## Abstract

Various nutrient recycling technologies are currently under development in order to alleviate the dependency of non-renewable raw material for the production of mineral phosphorus fertilizers commonly used in agriculture. The resulting products, such as struvites and ashes, need to be assessed for their application as so-called recycling-derived fertilizers (RDFs) in the agricultural sector prior to commercialization. Here, we conducted a short-term (54 days) trial to investigate the impact of different phosphorus fertilizers on plant growth and the soil P cycling microbiota. *Lolium perenne* was grown with application of superphosphate (SP) as inorganic fertilizer, two ashes (poultry litter ash (PLA) and sewage sludge ash (SSA)), and two struvites (municipal wastewater struvite (MWS) and commercial CrystalGreen^®^ (CGS)) applied at 20 and 60 kg P ha^−1^ in four replicates. A P-free control (SP0) was also included in the trial. Struvite application increased plant dry weights, and available P acid phosphatase activity was significantly improved for struvites at the high P application rate. The ash RDFs showed a liming effect at 60 kg P ha^−1^, and PLA60 negatively affected acid phosphatase activity, while PLA20 had significantly lower *phoD* copy numbers. P mobilization from phosphonates and phytates was not affected. TCP solubilization was negatively affected by mineral SP fertilizer application at both P concentrations. The bacterial (16S and *phoD*) communities were only marginally affected by the tested P fertilizers. Overall, struvites appeared to be a suitable substitute for superphosphate fertilization for Irish *L. perenne* pastures.

## 1. Introduction

Phosphorus (P) is present in all life. It is essential in the energy-carrying molecule ATP and in both DNA and RNA [1]. P is an essential macronutrient for plants and is therefore a crucial element for food production. Intensification of modern agriculture requires P fertilizer application to soil to compensate for the P uptake by plant harvests and retain soil productivity [2]. Current P management is facilitated in a linear manner, where P is extracted as phosphate rock from mines located outside Europe, with the largest abundance in Morocco and the Western Sahara, processed to mineral P fertilizers and applied to land. The excessive use of these fertilizers to increase crop productivity causes P runoff and leaching, one of the main causes of eutrophication of water bodies [3]. Additionally, the management of P-rich wastes, either from food production/agriculture, wastewater or industrial wastes are not optimal. The focus has been more on P removal than P recovery due to measures to limit environmental pollution; however, these practices need to shift towards a more sustainable approach [4,5]. P cannot be substituted by any other element, and it is becoming increasingly scarce. Its depletion as fossil resource, the main source for mineral P fertilizer production, has been predicted within the next few decades [6,7].

The soil microbiota, constituting of a plethora of bacteria and fungi, has an important role in nutrient cycling and overall soil structure and fertility [8,9]. Soil microbes are involved in organic matter decomposition, nutrient mineralization, and immobilization, and interactions between bacteria, fungi, and plants can be established to facilitate nutrient acquisition [10]. Increased microbial activity is usually witnessed in the soil rhizosphere, which describes the soil volume influenced by the plant root. During uptake of bioavailable orthophosphate by plant roots, a P depletion zone is created. The plants exude a range of compounds, such as amino acids, organic acids, sugars and vitamins, which influence the soil microbes habituating the soil rhizosphere, stimulating microbial nutrient mobilization activities from P pools that are inaccessible for plants [11]. While fertilizers can enhance crop production in the short term, their impact on the soil P cycle and the ability of the soil microbiota to replenish P in the long term can be seriously affected. Reductions in soil biodiversity and therefore a loss in functionality of the soil nutrient cycle have been reported upon long-term mineral P fertilizer application and change in land use [12,13,14,15,16].

In order to establish global food security and to avoid or reduce dependence upon unbalanced distributed resources located in geopolitically unstable countries and the ecological implications of environmental nutrient pollution, nutrient recovery from waste streams has been previously investigated in detail [17,18]. Potential sources for P recovery include wastewater, sewage sludge and manure. The resulting products of a multitude of P recovery processes are termed recycling-derived fertilizers (RDFs) or bio-based fertilizers [19,20]. While the manufacturing units used to produce RDFs are mostly at pilot-production scale, which makes their current production costly and uncompetitive compared to conventional mineral P fertilizers, these technologies might gain increased importance in the near future, when high P demand or military conflict in producer countries causes fertilizer price surges [21,22]. It was reported that sewage sludge disposal accounted for 40% of the greenhouse gas emissions of wastewater treatment plants. This could be significantly reduced by the adoption of a circular economy and enable sludge valorization for RDF production [23,24]. According to Ramankutty and colleagues [25], 70% of agricultural land use in the world comprises grassland. In Ireland, around 90% of its agriculture is dominated by grass-based systems [26]. Therefore, evaluation of these novel fertilizer products as alternatives to phosphate rock-based P fertilizers in grasslands is beneficial to support their widespread use in these agroecological systems.

In this experiment, four RDFs were evaluated regarding their P supply for perennial ryegrass (*Lolium perenne*) cultivation in comparison to superphosphate in pots. Two P concentrations of 20 and 60 kg P ha^−1^ were used for each fertilizer, and all fertilized treatments were compared to a P-free control. It was hypothesized that a) the RDFs would have similar fertilization value as conventional superphosphate at low and high P fertilization with comparable shoot biomass yield, nutrient uptake, and agronomic efficiency, and b) RDFs have a lower impact on the soil microbiota involved in P cycling. The objective was to analyze the abundance, composition, and function of the soil P-mobilizing bacteria under different P fertilizer products as well as no P addition and the resulting impact on grassland productivity.

## 2. Results

### 2.1. Shoot Biomass Yield, Agronomic Efficiency and Elemental Analysis of Ryegrass Biomass

Dry biomass yield of the *L. perenne* shoots ranged between 1.9 and 3.7 g per pot (Table 1). The average dry weight values of the high P fertilizer corrected for the seed germination rate (dry weight-to-shoot count ratio) treatments were higher than the ones with low P fertilization, with one exception: MWS20 yielded a higher dry weight-to-shoot count ratio than MWS60. The PLA20 treatment displayed the lowest values of all treatments, even performing poorer than the P-free control SP0. Generally, both struvites yielded the highest plant dry weight per shoot count on average, with MWS20 and CGS60 having significantly higher (*p* < 0.05) yields than the P-free control. The P utilization efficiency was evaluated as agronomic efficiency (P-uptake/P-supplied; AE). Because of its limited dry matter yield, averaging below SP0, the AE for PLA20 was negative. Interestingly, the low-P-input treatments SP20, SSA20 and MWS20 resulted in a higher AE than high-P-input equivalents in the short term.

The mass balance of the macro- (Table 2) and micronutrients (Table 3) could only be determined for the pooled treatments. Therefore, because only one measurement was obtained per treatment, statistical significance between the treatments could not be evaluated. Interestingly, Ca, K and N uptake was highest in PLA20, which was lowest in terms of dry biomass yield, and a similar pattern was visible for CGS20, which also exhibited low grass yields. Mg and S were similar among all treatments. P uptake was highest in SP60, and all treatment except for PLA showed higher P uptake in the high-P-fertilized treatments compared to the low-P-application rate of the same fertilizer.

The most abundant micronutrient was iron: its uptake was prominently higher in SP0 and PLA20, the treatments with the lowest dry biomass yield. Fe was followed by Mn, Zn and C in abundance, B and Cu were detected at particularly low concentrations.

### 2.2. RDF Influence on P-Mineralizing and P-Solubilizing Microbial Community

The MPN approach yielded no statistically significant difference in P-mobilization capabilities from phosphonoacetic acid (PAA), phytate, or overall heterotrophic bacteria (Table 4). Nonetheless, average MPN g^−1^ soil of PLA60, SSA60 and CGS60 was around twice as high or higher than for SP60 in terms of PAA mobilization. For the TCP solubilization, mineral P fertilizer had a lower CFU count than the RDFs. The SP60 treatment had the lowest CFU counts per gram soil and was also significantly lower (*p* < 0.05) than the RDFs PLA60, MWS20 and MWS60.

The pH-CaCl_2_ was significantly higher for all fertilized treatments compared to SP0 (Table 5). It is noteworthy that the pH value was higher for the higher P application rate of each fertilizer compared to the lower dose. The ash fertilizers PLA and SSA had a significantly higher liming effect on the soil compared to the other treatments at their high-P-application rate, followed by the mineral SP fertilizer and finally the struvite RDFs, which had the least effect on the soil pH. Morgan’s P test, on the other hand, revealed highest P availability in the struvite treatments; however, a significant difference was not reached due to high inter-sample variability. While the ash RDFs and the mineral fertilizers averaged in an index 1 range, the treatments MWS60 and CGS60 averaged in index 2 and index 3 range concentrations, respectively. The high variability in Morgan’s P results could stem from the RDF properties. While the ashes consisted of a fine powdery structure, the struvites were received as granules and therefore they were not as well distributed throughout the soil. This could have led to some soil samples containing more granules than other replicates, causing higher variability in the results due to higher P fertilizer input.

Potential acid phosphomonoesterase (ACP) activity was around 10-fold higher than alkaline phosphomonoesterase (ALP) activity in the rhizosphere soil measured after destructive harvest of the pots (Table 6). The mineral SP60 fertilizer and the struvite treatments MWS60 and CGS60 demonstrated a positive effect on potential ACP activity. Among the ash treatments, no significant differences from the P-free SP0 control were found, with the exception of SSA20, which had the highest measured activity of the ashes on average. The potential ALP activity showed no significant difference between treatments. While there was no significant difference detected in the *phoC* gene copy numbers, a significantly lower (*p* < 0.05) *phoD* gene copy number was found with the PLA20 treatment than the CGS20 treatment, which yielded the highest copy-number count for *phoD* (Table 6). In contrast to the potential enzyme assay, the *phoD* gene copy numbers were 1000-fold higher than *phoC*.

### 2.3. DNA Fingerprint Analysis of Bacterial Community Structure and Next-Generation Sequencing of Bacterial 16S rRNA Gene

PCR-DGGE was performed for individual treatments PLA20 and PLA60; SSA20 and SSA60; and MWS20, MWS60, CGS20 and CGS60, always in comparison to SP0 and SP60 to analyze the bacterial community structure. UPGMA dendrograms demonstrated high similarity between the band patterns of all treatments, and the similarity was always above 90% (data presented previously in [27]).

For 16S rRNA amplicon sequencing, a total of 7,954,388 demultiplexed sequences were obtained. On average, 180,782 sequences per sample were input into DADA2, 161,072 (86.1%) were filtered, 127,158 sequences per sample (66.3%) were merged and 74,522 (39.4%) of these were detected as non-chimeric. None of the alpha-diversity estimators were statistically significantly different. Chao1 values ranged from 864 (PLA60) to 1758 (SP0), while the Shannon index ranged from 5.6 (SSA20) to 7.1 (SP0). The richness estimators ACE, Chao1 and observed features were identical due to the removal of singletons in the ASV matrix output in QIIME2.

The CCA biplot of the 16S rRNA bacterial community analyzed via amplicon sequencing displayed limited separation of treatments (Figure 1). Of the treatments, only SP0 was separated on the first axis, while MWS60 was separated on the second axis. Pairwise comparison of treatments using the PERMANOVA as described in the pairwise Adonis function of the vegan package in R revealed no significant differences when Benjamini–Hochberg correction was applied. Significant environmental parameters involved in the visible shift of the bacterial soil community compared to the non-P-fertilized treatment were soil pH, P plant availability, acid phosphatase activity, TCP solubilizing CFUs and dry weight yields of *L. perenne*.

The phyla *Actinobacteria*, *Planctomycetes*, *Firmicutes*, *Proteobacteria*, *Verrucomicrobia* and *Acidobacteria* comprised over 75% of all phyla. The mean relative abundance of the 10 most abundant phyla detected in the 16S rRNA gene analysis can be seen in Appendix A.

Out of the 21 phyla of the 16S rRNA sequencing, five showed statistically significant differences in pairwise post hoc comparison. *Acidobacteria* were significantly more abundant (*p* < 0.05) in PLA60 in comparison to CGS20 (Figure 2). *Patescibacteria* were significantly more abundant (*p* < 0.05) in the ash treatment SSA60 and the struvite treatments MWS20 and CGS60 compared to PLA60 and SSA20 (Figure 2). *Euryarcheota*, on the other hand, were also significantly more abundant (*p* < 0.05) in MWS20 than in the SSA ash treatment at both concentrations (Figure 2). *Halanaerobiaeota* had significantly lower abundance (*p* < 0.05) in all P-fertilized treatments compared to the control SP0 (Figure 2). The phylum WS4 was significantly more abundant (*p* < 0.05) in the SSA60 treatment compared to SP20 and CGS60.

For the analysis of significant differences within the ASVs assigned down to genus level, genera with a relative abundance above 20% were selected for Kruskal–Wallis differential abundance analysis. Statistical analysis of all treatments together only revealed significant differences (*p* < 0.05) in the genus *Nocardioides* (SP20 was significantly different to SSA20). The differential abundance analysis was then repeated with subset of treatments, where SP0 was compared to all RDF20, SP0 to all RDF60, SP20 to all RDF20 and SP60 to all RDF60. For the comparisons SP0 vs. RDF20 and SP20 vs. RDF20, no significant differences were detected in the 20 most abundant genera for the 16S rRNA sequencing analysis. Differential abundance analysis of SP0 vs. RDF60 and SP60 vs. RDF60 revealed in both cases significant differences (*p* < 0.05) in the genera *Nocardioides*, *Conexibacter*, *Clostridium sensu stricto 13* and an uncultured bacterium belonging to the family *Isosphaeraceae.* The results for SP0 vs. RDF60 and SP60 vs. RDF60 Wilcoxon post hoc analyses were overlapping, with the SP60 vs. RDF60 having one additional significantly different (*p* < 0.05) treatment pair. Therefore, only the results for SP60 vs. RDF60 are shown, as summarized in Figure 3 below.

The uncultured *Isophaeraceae* had the highest relative abundance in the PLA60 treatment, while MWS60 demonstrated significantly lower (*p* < 0.05) abundance of this genus compared to PLA60. For the other three genera, MWS60 was significantly more abundant (*p* < 0.05) than most other treatments, while PLA60 was significantly less (*p* < 0.05) abundant than any other treatment. For the genus *Conexibacter* also, the mineral SP60 treatment showed significantly higher (*p* < 0.05) relative abundance in combination with MWS60, and the genus *Clostridium sensu stricto 13* was significantly more (*p* < 0.05) abundant in the MWS60 as well as the SSA60 treatment.

### 2.4. Next-Generation Sequencing Analysis of the phoD-Harboring Bacterial Community

A total of 4,570,420 *phoD* paired-end amplicon sequences were received from the sequencing facility, out of which 2,734,589 (59.8%) were successfully merged using USEARCH; 2,189,710 of the merged sequences (80.1%) were filtered and these sequences contained 1,260,580 (86.3%) singletons. In total, 9,204 OTUs were picked at a 75% similarity threshold and 3,278 chimeral sequences (1.6%) were removed. Finally, 2,356,148 (86.2%) out of 2,734,589 merged sequences were mapped to OTUs.

Alpha-diversity analysis via the phyloseq package in R showed no significant difference between the treatments for the estimates of observed features: Chao1 (5164–5659), ACE (5291–5878), Shannon (6.4–6.5) and Simpson (0.99). The community structure of *phoD*-harboring bacteria displayed in a CCA biplot (Figure 4) shows a visible separation of the MWS60 treatment from all other treatments on the first axis and a separation of the SSA60 treatment on the second axis. However, PERMANOVA using 999 permutations and subsequent pairwise comparison with Benjamini–Hochberg correction revealed no significant differences (*p* > 0.05) between the treatments.

The ten most abundant genera were *Bifidobacterium, Oerskovia, Frigoribacterium, Streptomyces*, *Xanthomonas*, *Microbacterium*, *Rhodopseudomonas*, *Leifsonia*, *Rubrivivax* and *Rhodanobacter*, and these genera constituted around 50% of the total genera in the samples (see Appendix A).

Six of the ten most abundant genera showed significant differences in their abundance in the post hoc analysis (*p* < 0.05, Figure 5). The genera *Oerskovia*, *Frigoribacterium*, *Xanthomonas* and *Rhodanobacter* were significantly less abundant in the PLA60 treatment. *Microbacterium* and *Rubrivivax*, on the other hand, had a significantly higher (*p* < 0.05) relative abundance in the PLA60 treatment. These genera had generally higher relative abundance in most ash treatments compared to the SP and struvite treatments MWS and CGS.

## 3. Discussion

The soil microbiota plays an important role in nutrient cycling. The transformation of inaccessible P compounds into plant-available orthophosphate is of particular interest. Their role in the nutrient cycle has received more attention in the past [9]. The use of biofertilizers, microbial inoculants with known N- and P-mobilization traits, has been under investigation for some time, and a meta-analysis conducted by Schütz and colleagues [28] found the greatest effect size for P-solubilizing microorganisms at a medium plant-available P concentrations of 25–35 kg ha^−1^ (Olsen-P); however, AMF performed better at lower plant-available P levels of 15–25 kg ha^−1^ and their inoculation increased plant yield by over 40%. The detrimental effects of mineral fertilizer application on the functioning of the nutrient cycling microbiota have been reported in several studies. Mozumder and Berrens [16] conducted a meta-analysis and described a significant correlation between the amount of inorganic fertilizer applied and biodiversity loss based on data collected across several countries. They, however, noted that information regarding specific fertilizer effects on soil biodiversity were lacking in the literature.

The influence of four different RDFs at two different P concentrations on the perennial ryegrass dry matter yield and the P-cycling soil microbiota was investigated in comparison to conventional mineral superphosphate fertilizer. At the beginning, it was hypothesized that the RDFs would provide a similar P fertilization effect as conventional mineral P fertilizer, while demonstrating less of an impact on the soil microbes involved in P-mobilization activities, i.e., share similarities with the no-P control.

In this study, all treatments except for poultry litter ash at the lower concentration (PLA20) showed improved *L. perenne* dry weight yields upon P fertilizer application, with one struvite (MWS60) producing significantly higher yields than all other treatments. This indicates that all RDF treatments contain plant-available P or provided suitable conditions for P to become plant-available. The agronomic efficiency was variable due to low overall dry matter yields, but presented its highest value for the MWS20 struvite. This is in contrast to findings in a study performed by González and colleagues [29] comparing the AE of several struvites produced from municipal wastewater to mineral triple15 fertilizer (N-P-K ratio 15:15:15). They found an increased AE for the triple15 treatment after 90 days of grass cultivation compared to the struvite treatments. However, the experiment was carried out in river sand and not in soil, limiting biological activity, and therefore microbial influence on P availability was lower compared to this study. Degryse and colleagues [30] argued that many studies use ground struvite mixed through soil to assess its fertilizing properties instead of granules, the latter of these likely having a slower dissolution rate. They also found that struvites dissolve much faster in acidic soils, with the dissolution rate dropping sharply from 0.43 mg per day to <0.05 mg per day for alkaline soils. In this study, the large CGS struvite granules were partially broken down, while the MWS struvite had a small particle diameter (not defined). This characteristic and the use of a rather acidic soil for this study could have caused an improved P release from struvite RDFs, as also reflected in higher dry matter yields.

P mobilization from phosphonate and phytate showed no significant fertilizer response, although some RDFs (PLA60, SSA60 and CGS60) displayed higher average MPNs for PAA utilization than the mineral fertilizer SP60. Furthermore, P solubilization was significantly affected in the SP60 treatment. This indicates a potential negative impact of high-P-mineral fertilizer on the P-cycling microbial community. Similarly, Fox and colleagues [31] reported significantly higher CFU for TCP-solubilizing bacteria, as well as phosphonate and phytate utilizing MPN in *Miscanthus giganteus* biochar-amended soils, although only compared to a P-free control treatment.

While potential acid phosphomonoesterase activity was increased for the treatments SP60, MWS60 and CGS60 in this study, it was lower in ash RDF treatments, likely due to the liming effect of the ashes, which was detected especially at higher P ash application rates. Though there are contradicting statements about the correlation between ACP and ALP activity and soil pH [32], it is generally assumed that ACP activity is higher at low soil pH and ALP activity at high soil pH levels [33,34]. However, next to soil pH, other soil properties, such as soil organic matter, moisture, clay and silt, total N, isotopically exchangeable P, and extractable Mg content, were significantly correlated with the intensity of phosphatase activity, according to a study conducted by Harrison [35]. In contrast to the findings in this study, Saha and colleagues [36] found increased ACP activity in soil fertilized with compost after three years of wheat/kidney bean/corn and okra crop rotation, while inorganic NPK fertilizer-treated soil demonstrated low activity. Interestingly, the ACP results in that study, investigating the impact of organic compost application on phosphatase activity, were 700–1500 µg p-NP g^−1^ soil, while ALP activities were roughly tenfold lower, similar to what had been recorded in this current study under RDF fertilization. This is likely due to the acidity of the soil, where ACP activity prevails. On a molecular level, however, *phoD* copy numbers were approximately 1000-fold more abundant than *phoC*. This indicates that the high ACP activity appears to be linked to plant root exudation activity rather than direct production of the enzyme by soil microbes [37].

Bacterial community structure sequencing analysis revealed differences in abundance upon different P fertilizer application in this study. *Acidobacteria* is a broadly abundant and highly phylogenetically diverse phylum that has been mostly investigated via cultivation-independent methods [38]. In the current study, a moderate trend of higher *Acidobacteria* relative abundance in the RDF ash treatments was noted, which is in contrast to findings by Jones and colleagues [39], who reported that the abundance of *Acidobacteria* was strongly regulated by soil pH and its abundance commonly higher at low soil pH levels. Alternatively, a significant positive correlation between *Acidobacteria* abundance and soil organic matter and Al content has been found [40]. Both ash RDFs contained high amounts of Al and PLA also contained mentionable amounts of total carbon, both of which could affect *Acidobacteria* abundance positively. Significantly positive correlations have been reported for relative abundance of *Patescibacteria* with plant-available P [41,42]. This phylum showed significantly higher abundances in two struvites as well as one ash RDF in the current study, and these treatments also exhibited high biomass yield. In terms of the CGS60 treatment, high available P was also obtained. Archaeal *Euryarchaeota* seemed to thrive on MWS20 struvite, while its abundance was lower for all other RDFs in the current study. Several genera of this phylum have been reported to have plant growth-promoting characteristics [43,44], but detailed research remains elusive [45]. The phylum *Halanaerobiaeota* was predominantly found in studies assessing compost, biochar and manure. Li and colleagues [46] reported *Halanaerobiaeota* as one of the dominant phyla present in chicken manure composting in combination with wood-derived biochar. Xie and colleagues [47] reported an increase in relative abundance of *Halanaerobiaeota* in the thermophilic phase of composting, and another study detected higher relative abundance of *Halanaerobiaeota* in biochar-supplemented anaerobic digestates [48]. These findings lead to the assumption that *Halanaerobiaeota* might be sensitive towards mineral P inputs, as the RDFs as well as the SP treatment contain forms of mineral P. At the genus level, then, *Nocardioides*, *Conexibacter* and *Clostridium sensu stricto 13* were negatively affected by the PLA60 treatment compared to all other high-P-fertilization treatments in this study. All three genera have been reported to participate in P-cycling activities [49,50,51].

Similarly to the bacterial 16S rRNA phylogenetic analysis, the evaluation of the *phoD*-harboring community did not reveal significant shifts in CCA, although some trends were observed in the form of a separation of MWS60 on the first and SSA60 on the second axis. Furthermore, alpha diversity of the *phoD*-harboring community was not significantly different. Differential abundant genera *Oerskovia*, *Frigoribacterium*, *Xanthomonas* and *Rhodanobacter* were significantly less abundant in PLA60, potentially impacting bacterial P mobilization originating from alkaline phosphomonoesterase. Of the most abundant genera identified in this current study, *Rhodanobacter*, *Streptomyces* and *Microbacterium* were also detected in the rhizosheath (sampling occurred <1 mm from the root) in a study assessing struvite fertilizer application on rhizosphere dynamics via 16S sequencing [52].

To summarize, struvite RDFs demonstrated improved ryegrass yields via higher P availability while simultaneously maintaining P-mobilization activities in soil. Mineral P application via superphosphate had detrimental effects on TCP-solubilizing bacteria. These findings partially support the hypotheses of improved P availability under an RDF regime, while struvites appear to have a lower impact on the P-cycling microbiota compared with superphosphate. The ash RDFs tested had more variable performance and some negative impacts on abundance of P-mobilizing genera. This was likely due to the ash’s different feedstocks, containing substances that could constrain soil bacteria. While short-term studies such as the present one are important to determine effects that can be observed at the first or second grass cut, long-term studies will also be necessary to establish the long-term effects of struvite and ash P fertilizers.

## 4. Materials and Methods

### 4.1. Pot Trial Setup and Fertilizer Preparation

The soil used in this experiment was a sandy loam (51% sand, 42% silt, 7% clay) with a pH (measured in 0.01 M CaCl_2_) of 5.0, a cation exchange capacity of 10.5 meq 100 g^−1^, organic matter content (determined via loss on ignition) of 5.3% and a Morgan’s P index of 2 (4.2 mg P L^−1^), indicating a likely crop response to the fertilizer application. The soil was taken from a field at Teagasc Johnstown Castle (Wexford, Ireland, N 52°17′47″, W 6°30′29″), which had been reseeded with perennial ryegrass in 2018 and prior to that was under long-term pasture management (personal communication with Patrick Forrestal). The soil was air-dried to a residual moisture content of 10–20% to allow for mixing, homogenizing, and sieving on one hand and to prevent damages to the soil microbes via complete dehydration on the other. Then, the soil was sieved through a mesh size of 5.60 mm to exclude bigger stones, followed by passing the soil through a smaller sieve (mesh size 3.35 mm) to remove smaller gravel and plant roots and to break down bigger soil aggregates. The moisture content and field capacity (water-holding capacity) of the sieved soil was then determined. Eleven treatments were prepared in quadruplicate. In detail, the RDFs poultry litter ash (PLA), sewage sludge ash (SSA), commercially available struvite (CrystalGreen^®^ Ostara, Richmond Heights, MO, USA; CGS), struvite from a municipal wastewater plant (MWS), and a P-free control (SP0) were used. The suitability of the RDFs was compared to a superphosphate (SP) fertilizer (SP40), which is a mixture of triple superphosphate and a filler (ratio 80:20) containing 16% P and 15% Ca and is commonly applied in grasslands in Ireland (personal communication with Patrick Forrestal). The SP and the struvite fertilizers exhibited different granule sizes, as visualized in Appendix A. Therefore, the larger granules were broken down (not ground) with a pestle and mortar to obtain smaller pieces of approximately 1 mm in diameter, thus potentially reducing bias by achieving higher accuracy in weighing of the products and improved similarity of distribution of the fertilizers when mixed with soil. The MWS fertilizer appeared to be moist, and was therefore dried in a fan oven at 40 °C until a weight equilibrium had been reached. The drying temperature for struvites did not exceed 40 °C to avoid N losses. P fertilization was carried out at two different concentrations: 20 kg ha^−1^, which is typical for grassland fertilization and also used for P buildup in soil and 60 kg ha^−1^, which is the recommended amount for pasture establishment [53]. The nutritional composition of the RDFs is displayed in Appendix A.

All fertilizers were applied to the soil in advance of setting up the pots, and the soil was thoroughly mixed upon fertilizer addition to avoid differences in microenvironmental conditions in the short-term trial. Additionally, because the amount of fertilizer was less than 0.5 g for the low P rate and less than 1.3 g for the high P rate per pot, the mixing of soil and fertilizer was carried out for each pot individually to ensure that every pot received the same amount of fertilizer. The mixture was applied in equal amounts of 825 g to the round plant pots (height = 10 cm, diameter_top_ = 12 cm, diameter_bottom_ = 9 cm) after a nylon mesh (20 µm mesh size, PlastOK, Birkenhead, UK) had been added to the bottom of the pot to minimize root growth outside the pot and reduce the loss of bulk soil from the pot. On top of the soil and fertilizer mix, a small layer of untreated soil was applied (approx. 3 cm) and the *Lolium perenne* (diploid var. AberGreen, provided by Germinal Ireland Ltd., Tipperary, Ireland) seeds were added to allow for similar seed germination conditions in all treatments without fertilizer contact. The quantity of seeds applied was derived from sowing recommendations received from Teagasc (Agricultural and Food Development Authority in Ireland) for a seed mixture of diploid (D) and tetraploid (T) varieties of *Lolium perenne* (40% AberGreen (D), 30% AberChoice (D) and 30% AberGain (T)) at a rate of 14 lb ac^−1^ (34.6 kg ha^−1^). The weight difference between the diploid and tetraploid seed mixture was corrected by counting and weighing 100 seeds of both the mixed varieties and the single variety in triplicates and using the weight ratio. Furthermore, the quantity of seeds was adjusted for the low germination rate of 17% in soil tested beforehand. The calculation of the seed quantity to be weighted in for each pot with all considerations mentioned above was carried out as follows (original calculation):$$\begin{array}{l} Seed\;application\;rate\;\left(\frac{kg}{pot}\right)\\ =Initial\;sowing\;rate\;\left(\frac{kg}{ha}\right)\;\times \;\frac{m_{mixed\;varieties}\left(kg\right)}{m_{single\;variety}\left(\;kg\right)}\;\times \;Germination\;rate\;\left(\%\right)\;\times \;Surface\;area\;\left(\frac{ha}{pot}\right) \end{array}$$

Finally, the pots were watered to obtain 70% of the water-holding capacity of the soil. The pots were placed on pot saucers with drains to prevent stagnant moisture. The trial was run for 54 days in a greenhouse. Irrigation of pots was carried out with rainwater collected at the Field Biological Unit at the University of Limerick every second day with an initial amount of 15 mL of water, which was gradually increased to 25 mL to account for plant needs. Two samples of the rainwater were taken throughout the course of the experiment to determine the nutrient composition and to confirm that the P intake from this source was negligible. The pots were weighed once a week to monitor the moisture status.

Shortly after seed germination, a full complement of the nutrients nitrogen (N, 220 kg ha^−1^), potassium (K, 185 kg ha^−1^), sulfur (S, 20 kg ha^−1^), magnesium (Mg, 30 kg ha^−1^), calcium (Ca, 440 kg ha^−1^), zinc (Zn, 5) and copper (Cu, 15 kg ha^−1^) was added to prevent starvation effects in the ryegrass due to lack of any nutrient other than P. The nutrient requirements for perennial ryegrass were obtained from Teagasc [54], as shown in Appendix A. The application of the nutrients in the form of solutions was conducted on two subsequent irrigation days instead of adding water. However, prior to the fertilizer solution application, 5 mL of water was added to precondition the soil. The solutions prepared from KNO_3_, Ca(NO_3_)_2_, MgCl_2_ and MgSO_4_ were added on the first occasion and CaCl_2_, CuSO_4_ and ZnSO_4_ were supplemented on the following irrigation day.

### 4.2. Harvest Procedure of Plants and Soil

The experiment was terminated after 54 days of perennial ryegrass cultivation. The pot irrigation was stopped five days in advance of the harvest. An overview of the experimental workflow can be found in Appendix A. The bulk soil of each pot was collected in one large bag per treatment by gently shaking off loosely attached soil after removal of the roots and soil from the pot. Then, the roots were put in ziplock bags, and the rhizosphere soil, which is closely attached to the plant roots, was shaken off more vigorously. The rhizosphere soil was stored at 4 °C immediately after harvest. Aliquots were taken for storage at −20 °C for DNA extraction at a later stage. The plant shoots were then cut off the roots, and the shoots were placed in paper bags to determine the fresh weight per pot and dry weight after drying for 72 h at 55 °C. Elemental analysis of the dried shoots was performed at Lancrop Laboratories Ltd. (Pocklington, UK), where atomic absorption spectroscopy, inductively coupled plasma spectrometry, spectrophotometry and titrations were employed according to ISO/IEC 17025:2005 to determine the boron, calcium, copper, iron, manganese, magnesium, molybdenum, nitrogen, phosphorus, potassium, sulfur and zinc content per pooled treatment.

### 4.3. Post-Harvest Experiments

The bulk soil was used to determine the soil pH for each sample. As stated previously [31], 5 g of air-dried and sieved soil was suspended in 20 mL of 0.01 M CaCl_2_ solution and rotated at 70 rpm for 5 min on an end-over-end RS-2M Intellimixer (ELMI, Riga, Latvia). After allowing the solution to settle for 2 h, the pH was measured potentiometrically in the supernatant. All other experiments were carried out with the rhizosphere soil.

For the MPN approach and the CFU analysis, a serial dilution of bacterial extracts from rhizosphere soil was carried out, as described previously [31]. There, 1 g of rhizosphere soil was weighed into a 50 mL assay tube, 10 mL of sterile saline solution (0.85% *w*/*v* NaCl, stored at 4 °C) was added aseptically and the suspension rotated at 75 rpm for 30 min at 4 °C, and 0.1 mL was taken immediately before the solution started settling and was applied seven times in a tenfold serial dilution in 0.9 mL sterile saline solution. Microtiter plates were filled with 180 μL MM2PAA medium, containing only phosphonate as single P source, MM2Phy medium, which contains only phytate (inositol hexaphosphoric acid) as single P source or R2 medium, a nutrient-reduced medium for heterotrophic growth [31,55]. Then, 20 μL of each dilution step (10^−1^ to 10^−7^) was applied in 5 technical replicates to the plates. The plates were closed with a lid, wrapped in parafilm to prevent evaporation, and incubated at 25 °C for 14 days. After 14 days of incubation the microtiter plate wells were assessed for cloudiness or color change indicating microbial growth. The number of positive wells where growth had been observed per dilution step was recorded in a table and a three-digit MPN value was derived using the dilution step with the highest dilution, where all wells were recorded as positive, and the number of positive wells of the two following higher dilutions. This MPN value was then compared to an MPN table provided by the Food and Drug Administration (FDA, Blodgett, https://www.fda.gov/food/laboratory-methods-food/bam-appendix-2-most-probable-number-serial-dilutions, accessed on 28 March 2023) to obtain the MPN g^−1^ (see Appendix A for further calculation details).

For the selected dilutions (10^−3^ and 10^−4^), 100 μL was applied to an individual TCP plate [31] in three technical replicates. The plates were then sealed with parafilm and were incubated at 25 °C in the dark for 14 days. CFU g^−1^ soil calculations are provided in Appendix A.

The potential acid and alkaline phosphomonoesterase activity in soil was determined following standard protocols [56] using 1 g of soil and a spectrophotometric method based on p-nitrophenyl phosphate. Detailed information is provided in theAppendix A.

The available P was measured using Morgan’s P test, which is typically employed for examining Irish soil P status [57], as described previously by Beech and English [58] and Murphy and Riley [59]. Details are provided in Appendix A. The results were compared to the soil P index for grasslands [54]. The P utilization efficiency (P-uptake/P-supplied; AE) [60] was evaluated for agronomic efficiency, as other means of evaluation were not possible due to the low dry matter yield.

DNA extraction from 0.25 g rhizosphere soil (frozen at −80 °C) was performed using the DNeasy PowerSoil Pro kit (QIAGEN GmbH, Hilden, Germany) according to the manufacturer’s instructions. Quantification of DNA extracts was carried out using the Qubit Fluorometer (Life Technologies, Carlsbad, CA, USA) with a Qubit dsDNA HS assay kit (Life Technologies, Carlsbad, CA, USA). 16S rRNA PCR for denaturing gradient gel electrophoresis (DGGE) was carried out with the primer pair 341F-GC and 518R developed by Muyzer, de Waal, and Uitterlinden [61], which targets a 233 bp-long gene fragment of the V3 region of the bacterial 16S rDNA gene. The PCR-DGGE was conducted as described previously [31,62], and a detailed description is provided in Appendix A.

Amplification of the V4 region of the 16S rRNA gene was carried out in triplicate with primers containing Illumina adapters (515F-Illumina 5′- TCG TCG GCA GCG TCA GAT GTG TAT AAG AGA CAG GTG CCA GCM GCC GCG GTA A-3′ and 806R-Illumina 5′- GTC TCG TGG GCT CGG AGA TGT GTA TAA GAG ACA GGG ACT ACH VGG GTW TCT AAT-3′). Each 25 µL reaction contained 1× KAPA HiFi HotStart ReadyMix (2.5 mM MgCl_2_, Roche, Basel, Switzerland), 0.3 µM of each primer and 0.5 µL DNA template and the PCR conditions were as follows: initial preincubation at 95 °C for 3 min, followed by 25 cycles of denaturation at 98 °C for 20 s, annealing at 65 °C for 15 s and elongation at 72 °C for 15 s, with a final extension step at 72 °C for 60 s. The PCR products were purified using the GenElute PCR Clean-up kit (Sigma Aldrich, St. Louis, MO, USA) according to the instruction manual, eluted in 50 µL, and then 20 µL was sent to CIBIO-InBIO GenomePortugal Research Infrastructure (Vairão, Portugal), where libraries were prepared. Illumina paired-end next-generation sequencing (NGS) was then performed using a 500-cycle Rapid Run kit (Illumina, San Diego, CA, USA) on a Hiseq2500 sequencer operated by Genewiz (Leipzig, Germany). Demultiplexing was also performed by the sequencing provider.

Sequencing of the functional *phoD* gene fragment was carried out at the University of Minnesota Genomics Centre (UMGC, Minneapolis, MN, USA) as a PE300 MiSeq amplicon sequencing run. First, the *phoD* gene fragment was amplified with the *phoD* primers *phoD*-F733 (5′-TGG GAY GAT CAY GAR GT-3′) and *phoD*-R1083 (5′-CTG SGC SAK SAC RTT CCA-3′) [63], which provide a high *phoD* coverage and diversity. The KAPA2G Robust HotStart PCR kit (KAPA Biosystems, Roche) was used and each 25 µL reaction contained 1× KAPA2G Buffer A, 1 M betaine, 0.5 mM MgCl_2_, 0.2 mM dNTPs, 0.8 mM of each primer 0.5 U KAPA2G Robust HotStart DNA Polymerase and 0.5 µL DNA template. The reaction conditions were as follows: 3 min of initial denaturation at 95 °C, followed by 35 cycles of denaturation at 95 °C for 30 s, annealing at 58 °C for 15 s, elongation at 72 °C for 15 s and a final extension step at 72 °C for 3 min. PCR products were purified using the GenElute PCR cleanup kit (Sigma Aldrich, St. Louis, MO, USA) according to the instruction manual. The triplicates were pooled and diluted 1:5 with PCR water and then subjected to a second PCR adding the Illumina adapter sequences. In this PCR, the KAPA HiFi HotStart Readymix (KAPA Biosystems, Roche) was used to amplify the samples in triplicate. Each 25 µL reaction contained 1× KAPA HiFi HotStart Readymix, 0.5 mM MgCl_2_, 0.4 mM of each primer and 1 µL of each template. The reaction conditions were 3 min initial denaturation at 95 °C, followed by 15 cycles of denaturation at 95 °C for 20 s, annealing at 65 °C for 15 s and elongation at 72 °C for 15 s, and a final extension at 72 °C for 1 min. The PCR products were pooled, purified with the GenElute PCR cleanup kit (Sigma Aldrich) again, and quantified using a Qubit Fluorometer (Life Technologies, Carlsbad, CA, USA) with a Qubit dsDNA HS assay kit (Life Technologies, Carlsbad, CA, USA) before sending the samples to UMGC for library preparation and sequencing and subsequent demultiplexing of sequences.

*phoC* qPCR was conducted with the primers *phoC*-A-F1 (5′-CGG CTC CTA TCC GTC CGG-3′) and *phoC*-A-R1 (5′-CAA CAT CGC TTT GCC AGT G-3′) reported by Fraser and colleagues [64] on a Roche LightCycler^®^ 96 (Roche Diagnostics, Mannheim, Germany) using the KAPA SYBR FAST qPCR Master Mix (KAPA Biosystems, Cape Town, South Africa). Each 10 µL reaction contained 1× KAPA SYBR FAST qPCR Master Mix (2.5 mM MgCl_2_), 0.3 mM of each primer and 1 µL of DNA template (diluted to 20 ng µL^−1^). The qPCR program started with an initial preincubation at 95 °C for 5 min, then 45 cycles of denaturation at 95 °C for 3 s, primer annealing at 60 °C for 20 s and elongation at 72 °C for 10 s followed. *phoD* qPCR was carried out using the ALPS primers ALPS-F730 (5′-CAG TGG GAC GAC CAC GAG GT-3′) and ALPS-R1101 (5′-GAG GCC GAT CGG CAT GTC G-3′) developed by Sakurai and colleagues [65], also applying the KAPA SYBR FAST qPCR Master Mix (KAPA Biosystems) on the LightCycler^®^ 96 (Roche). Each 10 µL reaction contained 1× KAPA SYBR FAST qPCR Master Mix (2.5 mM MgCl_2_), 0.3 mM of each primer and 1 µL of DNA template (diluted to 20 ng µL^−1^). The qPCR program was set up as follows: 5 min preincubation at 95 °C, followed by 45 cycles of denaturation at 95 °C for 3 s, annealing at 60 °C for 20 s, and extension and data acquisition at 72 °C for 20 s. The values obtained for *phoC* and *phoD* qPCR were then transformed into copies per gram of soil in order to normalize for the DNA extraction yield.

### 4.4. Statistical Analyses

The results obtained from the shoot dry weight, pH measurements, MPN and CFU analysis, soil ACP and ALP enzymatic assay, Morgan’s P test, *phoD* and *phoC* gene fragment quantification were analyzed for statistical significance with the SPSS software (SPSS Statistics, IBM, Version 26). Because the number of samples was fewer than 50, normality was checked using the Shapiro–Wilk test at *p* > 0.05. The homogeneity of variance was evaluated using Levene’s test at *p* > 0.05, and if both assumptions were fulfilled, a one-way analysis of variance (ANOVA) was performed, applying the Tukey HSD post hoc test for pairwise comparison of the treatments (*p* < 0.05) to assess significant differences between treatment means. Data violating one or both assumptions of normality and homogeneity of variance were transformed either using logarithm or square root transformation, then the ANOVA was repeated, and the untransformed results were reported. When Levene’s test gave results with *p* < 0.05, the Games–Howell test was applied instead of the Tukey post hoc analysis. If normality of the data was not achieved, a non-parametric Kruskal–Wallis test was performed instead.

DGGE gel images were compared in Phoretix 1D software (Totallab, Newcastle, UK), creating unweighted pair-group method with arithmetic mean (UPGMA) dendrograms. Binary matrices were used for a canonical correspondence analysis (CCA), testing effects of measured environmental factors and visualizing the CCA biplot using the R packages vegan, mabund, permute and lattice, applying a permutational multivariate analysis of variance (PERMANOVA) test using 999 permutations (R Studio, Version 4.0.3).

Demultiplexed, paired-end 16S rRNA gene sequence reads obtained from CIBIO-InBIO were imported into QIIME2 2020.8 [66]. The paired-end reads were joined, quality filtered and denoised via the q2-dada2 plugin [67]. The resulting amplicon sequence variants (ASVs) were aligned using the q2-alignment with mafft [68] and a phylogenetic tree was built via q2-phylogeny with fasttree2 [69]. Samples were rarefied to 14,211 sequences per sample and the alpha-diversity metrics (observed species and Faith’s phylogenetic diversity [70] and beta diversity metrics (Bray–Curtis dissimilarity, Jaccard distance, unweighted UniFrac [71] and weighted UniFrac [72]), and principal coordinate analysis (PCoA) were assessed using the q2-diversity plugin. Taxonomy assignment was accomplished using the q2-feature-classifier [73], training with a SILVA 13.8 99% reference data set for 16S rRNA [74,75]. First, the reads in the reference data set were trimmed to the region of interest using the primers that had been used in PCR amplification prior to the sequencing. Then, a naïve Bayes classifier was trained with the trimmed reference sequences and their reference taxonomies. The mvabund and vegan packages in R software (version 4.0.3) were used to perform a CCA analysis based on the ASV table obtained during the QIIME2 analysis, and the results in the ASV table were visualized as CCA. Additionally, taxa bar plots were created to disclose the percentage relative abundance distribution of prevalent taxa among the treatments. Differential abundance was tested using the Kruskal–Wallis rank-sum test with the function kruskal.test in R Studio (version 4.0.3) and Wilcoxon post hoc analysis for multiple pairwise comparisons between groups (function pairwise.wilcox.test), applying Benjamini–Hochberg correction for multiple testing.

Raw *phoD* sequences obtained using the Illumina platform were made available demultiplexed by the University of Minnesota Genomics Center. Primers and poor-quality sequences were removed using cutadapt software [76]. Usearch [77] was used to merge paired-end reads of the PE300 *phoD* amplicon data. Then, the reads were filtered via the fastq_filter command, unique sequences selected, duplicates removed via the fastx_uniques command, and the UPARSE pipeline used to cluster sequences into centroid operational taxonomic units (OTUs) [78]. A 75% sequence similarity threshold was used for OTU clustering [79]. A reference database for the alkaline phosphomonoesterase gene *phoD* was downloaded from the FunGene functional gene repository [80] and was used as the database to assign taxonomy. The OTU table was not rarefied to avoid omission of rare OTUs and decreased sensitivity due to an increase in type II errors [81]. The mvabund and vegan packages in the R software (version 4.0.3) were used to create a CCA biplot and subsequent permutation based on the OTU table obtained during the Usearch analysis. Additionally, taxa bar plots were created to disclose the percentage relative abundance distribution of prevalent genera among the treatments, as described above for the 16S NGS data.

## 5. Conclusions

At the outset of this study, we hypothesized that struvites and ashes would perform similarly to superphosphate in terms of yield, nutrient uptake and agronomic efficiency. Application of struvite has demonstrated the potential to replace superphosphate in Irish grasslands and potentially further afield where soil conditions are comparable. In contrast, the performance of ashes was in part below that of superphosphate and struvites in the short-term. The use of ashes may be of interest for long-term agronomic benefits, though, and this warrants further investigations.

The overall microbiological parameters revealed stable communities that were only marginally affected by the different P fertilizers. Thus, at least in the short-term, fertilization with superphosphate, struvites or ashes had no dramatic effect on the bacterial communities. Nevertheless, some microbiological parameters suggested that superphosphate and ashes have a negative influence on bacterial functions of P cycling. This observation has been made possible through a combination of state-of-the-art cultivation-based and cultivation-independent forms of analysis.

While further research is recommended on RDF use in grasslands to determine long-term effects of struvites and especially ashes, we concluded that struvites in the current study were the best option for sustainable pasture fertilization, performing best at plant and microbe level.

## Figures and Tables

**Figure 1 plants-12-02762-f001:**
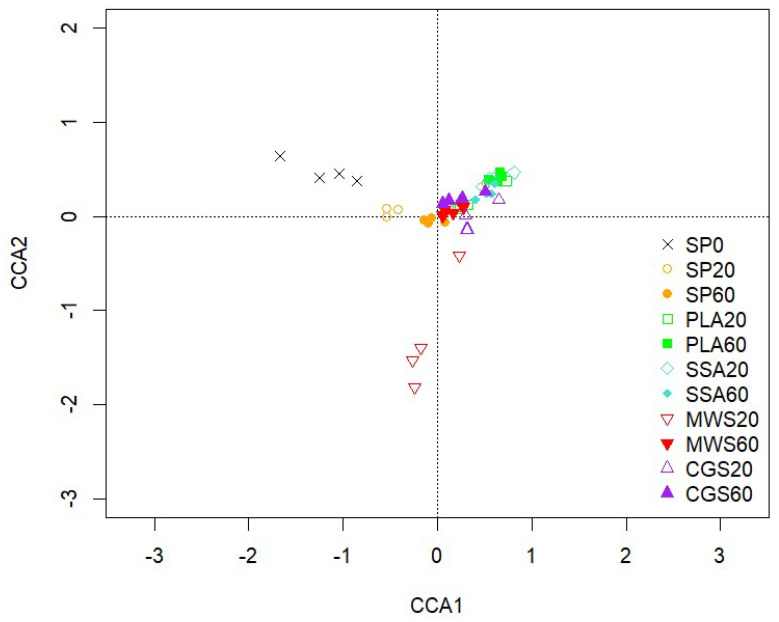
CCA of the bacterial community sequences based on ASVs obtained from 16S rRNA, CCA1 explains 3.44% and CCA2 3.05% of the total variation of the data, n = 4.

**Figure 2 plants-12-02762-f002:**
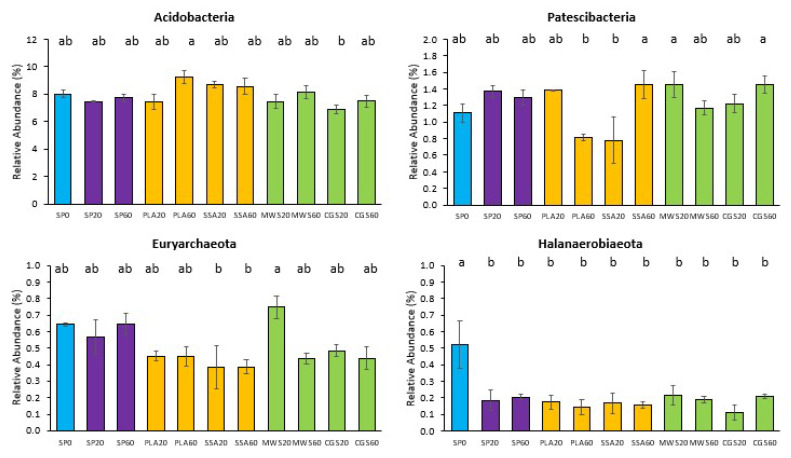
Mean relative abundance of selected phyla of the 16S rRNA sequencing with an abundance above a cutoff of 1.5% that showed significant differences in the treatments (SP0 = control, blue; SP20/40 = superphosphate, purple; PLA/SSA 20/40 ashes = yellow; MWS/CGS 40/60 struvites = green). Significance determined via Kruskal–Wallis test and Wilcoxon post hoc analysis with Benjamini–Hochberg correction, different letters indicate significant differences (*p* < 0.05), n = 4.

**Figure 3 plants-12-02762-f003:**
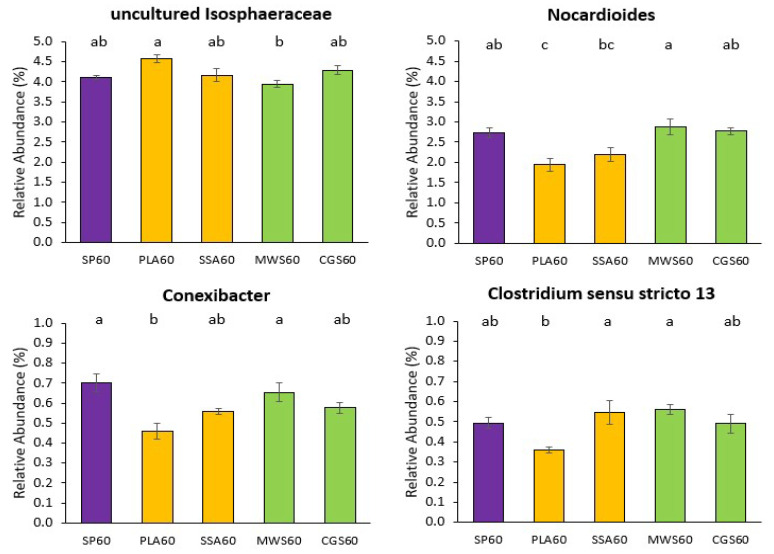
Significant differences in mean relative abundance of selected genera above the cutoff of 20% overall relative abundance on 16S rRNA sequencing analysis for the high-P treatments SP60 (purple), PLA60 & SSA60 (yellow), MWS60 & CGS60 (green) only. Significance determined via Kruskal–Wallis test and Wilcoxon post hoc analysis with Benjamini–Hochberg correction, different letters indicate significant difference, n = 4.

**Figure 4 plants-12-02762-f004:**
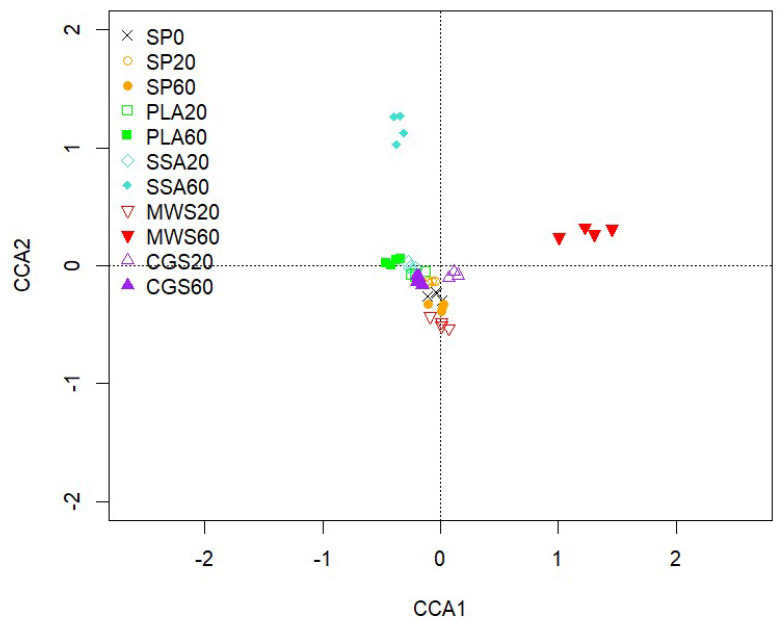
CCA of the *phoD* gene-harboring community sequences based on OTUs. CCA1 explains 3.04% and CCA2 2.73% of the total variation of the data, n = 4.

**Figure 5 plants-12-02762-f005:**
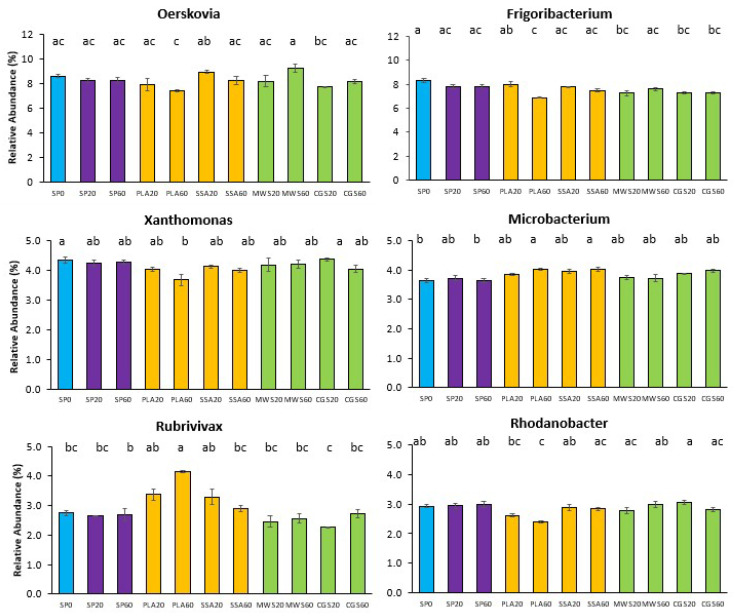
Relative abundance (%) bar plots of top 10 genera with significantly different abundance on *phoD* sequencing analysis (SP0 = control, blue; SP20/40 = superphosphate, purple; PLA/SSA 20/40 ashes = yellow; MWS/CGS 40/60 struvites = green). Significance determined via Kruskal–Wallis test and Wilcoxon post hoc analysis with Benjamini–Hochberg correction, different letters indicate significant differences, n = 4.

**Table 1 plants-12-02762-t001:** Average values for fresh weight (FW) and dry weight (DW) of *Lolium perenne* harvest, shoot count, and dry weight-to-shoot ratio, measured in quadruplicate after 54 days of growth in pots, and agronomic efficiency (AE). Different letters indicate significant differences (*p* < 0.05) within a column, determined via one-way ANOVA with Tukey HSD or Games–Howell post hoc analysis, ± represents standard error.

Treatment	Biomass DW (g)		Shoot Count		DW:Shoot Ratio (mg Shoot^−1^)		AE
SP0	2.29 ^c^	±0.06	121.25 ^a^	±1.9	18.89 ^bc^	±0.5	
SP20	2.66 ^ac^	±0.10	127.25 ^a^	±3.6	20.88 ^ac^	±0.5	12.50
SP60	2.52 ^ac^	±0.05	109.75 ^a^	±4.1	23.04 ^ab^	±1.0	2.73
PLA20	2.14 ^c^	±0.16	118.50 ^a^	±4.7	18.19 ^c^	±1.4	−8.74
PLA60	2.72 ^ac^	±0.16	117.25 ^a^	±9.1	23.40 ^ab^	±1.3	6.60
SSA20	2.59 ^ac^	±0.06	125.00 ^a^	±5.7	20.79 ^ac^	±0.8	14.38
SSA60	2.75 ^ac^	±0.17	122.00 ^a^	±4.5	22.64 ^ac^	±1.7	7.15
MWS20	3.19 ^ac^	±0.18	117.25 ^a^	±10.9	27.69 ^a^	±2.1	23.89
MWS60	2.79 ^ac^	±0.34	107.50 ^a^	±6.0	26.07 ^ab^	±3.2	13.31
CGS20	2.27 ^ac^	±0.15	102.25 ^a^	±7.7	22.24 ^ac^	±0.6	2.21
CGS60	2.85 ^ab^	±0.09	110.25 ^a^	±8.1	26.36 ^a^	±2.5	8.85

**Table 2 plants-12-02762-t002:** Mass balance of total perennial ryegrass shoot dry weight (pooled per treatment), in µg per treatment (P added as 0, 20 or 60 kg ha^−1^), for the macronutrients of the pot trial after 54 days of growth in a greenhouse.

Treatment	Macronutrients (µg Treatment^−1^)					
	Calcium	Magnesium	Sulphur	Nitrogen	Phosphorus	Potassium
SP0	1619.8	371.2	360.0	4713.2	258.7	4836.9
SP20	1397.2	329.3	339.3	4760.5	299.4	4471.1
SP60	1360.3	321.9	363.4	5077.9	415.4	4911.7
PLA20	1777.8	407.4	419.8	5530.9	296.3	5370.4
PLA60	1245.3	290.3	318.4	4307.1	252.8	4194.8
SSA20	1275.8	294.4	323.8	4367.0	235.5	4210.0
SSA60	1135.7	267.8	313.9	4478.3	295.5	3942.8
MWS20	1194.6	280.5	298.6	4461.5	262.4	4181.0
MWS60	944.0	272.0	264.0	3944.0	304.0	3576.0
CGS20	1507.2	363.0	374.0	5181.5	308.0	5005.5
CGS60	992.0	301.2	292.3	4242.7	336.6	3950.4

**Table 3 plants-12-02762-t003:** Mass balance of total perennial ryegrass shoot dry weight (pooled per treatment), in µg per treatment (P added at 0, 20 or 60 kg P ha^−1^), for the micronutrients of the pot trial after 54 days of growth in a greenhouse.

Treatment	Micronutrients					
	Boron	Copper	Iron	Manganese	Molybdenum	Zinc
SP0	0.9	5.5	217.8	36.2	0.01	6.5
SP20	0.6	3.3	116.8	30.7	0.02	5.8
SP60	0.6	3.1	151.7	27.7	0.02	6.2
PLA20	1.0	6.1	234.0	32.8	0.02	6.6
PLA60	0.6	2.7	146.3	18.5	0.01	5.3
SSA20	0.7	3.5	193.8	23.9	0.01	5.1
SSA60	0.6	3.0	139.8	17.1	0.01	5.4
MWS20	0.6	2.9	144.1	24.1	0.01	4.3
MWS60	0.4	2.3	102.0	21.0	0.01	3.8
CGS20	0.7	2.9	158.0	30.6	0.01	5.5
CGS60	0.5	2.9	122.0	23.6	0.01	4.8

**Table 4 plants-12-02762-t004:** MPN values of phosphonoacetic acid (PAA) mobilizing and phytate (Phy) mobilizing bacteria and MPN values of total heterotrophic bacteria (R2A) and CFU values of tricalcium phosphate solubilizing bacteria (TCP). Different letters indicate significant differences (*p* < 0.05) within a column, determined via one-way ANOVA with Tukey HSD or Games–Howell post hoc analysis, ± represents standard error, n = 4.

Treatment	PAA (MPN g^−1^ Soil)		Phy (MPN g^−1^ Soil)		R2A (MPN g^−1^ Soil)		TCP (CFU g^−1^ Soil)	
SP0	3.1 × 10^6 a^	±1.0 × 10^6^	18.6 × 10^6 a^	±5.6 × 10^6^	92.5 × 10^6 a^	±2.5 × 10^7^	275.0 × 10^3 bcd^	±3.1 × 10^4^
SP20	7.5 × 10^6 a^	±3.0 × 10^6^	11.5 × 10^6 a^	±3.5 × 10^6^	141.3 × 10^6 a^	±3.5 × 10^7^	150.0 × 10^3 cd^	±3.2 × 10^4^
SP60	6.7 × 10^6 a^	±1.2 × 10^5^	8.3 × 10^6 a^	±2.9 × 10^6^	120.0 × 10^6 a^	±1.9 × 10^7^	138.0 × 10^3 d^	±3.6 × 10^4^
PLA20	9.8 × 10^6 a^	±2.6 × 10^6^	19.8 × 10^6 a^	±7.0 × 10^6^	218.8 × 10^6 a^	±5.1 × 10^7^	250.0 × 10^3 bcd^	±5.0 × 10^4^
PLA60	14.2 × 10^6 a^	±5.9 × 10^6^	11.3 × 10^6 a^	±1.2 × 10^5^	102.5 × 10^6 a^	±2.4 × 10^7^	1.0 × 10^6 a^	±7.6 × 10^5^
SSA20	5.2 × 10^6 a^	±7.4 × 10^5^	31.3 × 10^6 a^	±1.3 × 10^7^	154.5 × 10^6 a^	±4.9 × 10^7^	850.0 × 10^3 ad^	±1.6 × 10^5^
SSA60	10.1 × 10^6 a^	±4.4 × 10^6^	12.8 × 10^6 a^	±9.4 × 10^5^	156.3 × 10^6 a^	±3.4 × 10^7^	488.0 × 10^3 ad^	±4.4 × 10^5^
MWS20	5.4 × 10^6 a^	±6.5 × 10^5^	19.3 × 10^6 a^	±3.2 × 10^6^	156.3 × 10^6 a^	±2.7 × 10^7^	925.0 × 10^3 ab^	±8.4 × 10^4^
MWS60	9.5 × 10^6 a^	±4.5 × 10^6^	16.0 × 10^6 a^	±2.0 × 10^5^	132.5 × 10^6 a^	±2.0 × 10^7^	863.0 × 10^3 ac^	±9.7 × 10^4^
CGS20	6.2 × 10^6 a^	±2.4 × 10^5^	19.8 × 10^6 a^	±6.7 × 10^6^	102.5 × 10^6 a^	±2.4 × 10^7^	575.0 × 10^3 ad^	±9.4 × 10^4^
CGS60	12.0 × 10^6 a^	±1.7 × 10^6^	15.3 × 10^6 a^	±1.3 × 10^6^	61.7 × 10^6 a^	±2.4 × 10^6^	763.0 × 10^3 ad^	±8.7 × 10^4^

**Table 5 plants-12-02762-t005:** Mean values for soil pH and bioavailable P measured using Morgan’s P test after 54 days of growth. Different letters indicate significant differences (*p* < 0.05) within a column, determined via one-way ANOVA with Tukey HSD or Games–Howell post hoc analysis, ± represents standard error, n = 4.

Treatment	Soil pH		Morgan’s P (mg P L^−1^)	
SP0	4.75 ^d^	±0.03	0.58 ^c^	±0.08
SP20	4.94 ^bc^	±0.02	1.01 ^ac^	±0.12
SP60	5.05 ^b^	±0.04	2.35 ^ab^	±0.26
PLA20	4.95 ^bc^	±0.02	1.98 ^ab^	±0.17
PLA60	5.22 ^a^	±0.03	2.75 ^b^	±0.19
SSA20	4.96 ^bc^	±0.02	1.57 ^ab^	±0.11
SSA60	5.24 ^a^	±0.02	2.48 ^ac^	±0.41
MWS20	4.87 ^cd^	±0.04	2.00 ^abc^	±0.33
MWS60	5.02 ^b^	±0.02	4.46 ^abc^	±0.60
CGS20	4.90 ^c^	±0.02	2.79 ^abc^	±0.45
CGS60	4.95 ^bc^	±0.01	5.16 ^abc^	±0.71

**Table 6 plants-12-02762-t006:** Average potential acid (ACP) and alkaline phosphomonoesterase (ALP) activity determined via spectrophotometry and *phoC* and *phoD* gene copy numbers per gram of soil normalized for DNA extraction yield determined via qPCR after destructive harvest of pot after 54 days of growth. Different letters indicate significant differences (*p* < 0.05) within a column, determined via one-way ANOVA with Tukey HSD or Games–Howell post hoc analysis, ± represents standard error, n = 4.

Treatment	ACP (µg pNP g^−1^ Soil h^−1^)		ALP (µg pNP g^−1^ Soil h^−1^)		*phoC* (*phoC* Copies g^−1^ Soil)		*phoD* (*phoD* Copies g^−1^ Soil)	
SP0	1653.7 ^d^	± 77.4	218.1 ^a^	± 10.6	131.3 × 10^3 a^	±6.3 × 10^4^	148.2 × 10^6 ab^	±5.1 × 10^6^
SP20	1663.9 ^d^	± 38.9	278.7 ^a^	± 10.2	79.7 × 10^3 a^	±3.3 × 10^4^	140.9 × 10^6 ab^	±2.9 × 10^6^
SP60	2167.0 ^abc^	± 78.8	259.5 ^a^	± 16.4	98.9 × 10^3 a^	±3.4 × 10^4^	140.7 × 10^6 ab^	±3.1 × 10^6^
PLA20	2007.1 ^bd^	± 45.6	208.4 ^a^	± 20.7	75.8 × 10^3 a^	±3.3 × 10^4^	131.7 × 10^6 b^	±7.0 × 10^6^
PLA60	1712.3 ^cd^	± 81.7	217.5 ^a^	± 7.0	119.9 × 10^3 a^	±5.9 × 10^4^	151.7 × 10^6 ab^	±4.0 × 10^6^
SSA20	2183.4 ^abc^	± 94.3	188.0 ^a^	± 7.1	75.9 × 10^3 a^	±2.9 × 10^4^	133.5 × 10^6 ab^	±6.9 × 10^6^
SSA60	2084.2 ^bd^	± 103.5	228.8 ^a^	± 27.2	88.3 × 10^3 a^	±4.5 × 10^4^	138.4 × 10^6 ab^	±6.6 × 10^6^
MWS20	2262.8 ^bd^	± 137.6	233.9 ^a^	± 33.3	100.8 × 10^3 a^	±3.6 × 10^4^	145.5 × 10^6 ab^	±1.7 × 10^7^
MWS60	2262.8 ^ab^	± 103.1	233.9 ^a^	± 30.4	168.6 × 10^3 a^	±7.5 × 10^4^	163.3 × 10^6 ab^	±5.3 × 10^6^
CGS20	2066.9 ^bd^	± 133.7	199.9 ^a^	± 23.2	218.4 × 10^3 a^	±5.0 × 10^4^	168.9 × 10^6 a^	±9.8 × 10^6^
CGS60	2616.4 ^a^	± 163.7	209.0 ^a^	± 17.9	88.2 × 10^3 a^	±3.3 × 10^4^	154.3 × 10^6 ab^	±4.1 × 10^6^

## Data Availability

DNA sequence data are deposited in NCBI’s Sequence Research Archive under the BioProject ID PRJNA990651 (accession numbers SAMN36271156 to SAMN36271199 for 16S and SAMN36272326 to SAMN36272369 for *phoD*).

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
