# Peer review of "Short-Term Impact of Recycling-Derived Fertilizers on Their P Supply for Perennial Ryegrass (Lolium perenne)"

_plants, 2023, doi:10.3390/plants12152762_

Round 1
Reviewer 1 Report
The paper is of interest in the field of research approached, but some improvements are suggested.
Title: please replace ”Their” from title with ”their”
1. Introduction
The introduction is well documented and updates the approached issue.
2.Results
When authors stated the objectives of the work, they made no mention about ”Shoot Biomass Yield, Agronomic Efficiency and Elemental Analysis of Ryegrass Biomass”, and the first issue approached in discussions is the above mentioned.
We suggest the authors include this issue in the presentation of the objectives.
Line 106. Please mention the source of the formula of agronomic efficiency of P utilization; if original, please mention. We consider that a better approach is to mention the formula in ”Material and Methods” section.
3.Discussion
The discussions complement the results and are edifying regarding the issue addressed.
4.Materials and Methods
Lines 478-482. Please mention the source of the formula; if original, please mention.
5.Conclusion
The conclusions are superficial. They do not fully explain how the objectives of the study were achieved. We suggest the authors to approach each part of the stated objective, formulated as we recommend.
We recommend authors to mention the novelty brought by your research compared to the present level of knowledge in the field and underline the originality of their work.
Author Response
Title: please replace ”Their” from title with ”their”
This has now been corrected.
- Introduction
The introduction is well documented and updates the approached issue.
Thank you.
2.Results
When authors stated the objectives of the work, they made no mention about ”Shoot Biomass Yield, Agronomic Efficiency and Elemental Analysis of Ryegrass Biomass”, and the first issue approached in discussions is the above mentioned.
We suggest the authors include this issue in the presentation of the objectives.
Thank you for the suggestion. We have now expanded the hypotheses to include these factors as suggested.
Line 106. Please mention the source of the formula of agronomic efficiency of P utilization; if original, please mention. We consider that a better approach is to mention the formula in ”Material and Methods” section.
Thank you for the suggestion. We have now partially moved this sentence to the materials and methods section and included a reference for phosphorus use efficiency (L559-61).
3.Discussion
The discussions complement the results and are edifying regarding the issue addressed.
Thank you for the comment.
4.Materials and Methods
Lines 478-482. Please mention the source of the formula; if original, please mention.
The formula was established independently, and this has now been highlighted as requested.
5.Conclusion
The conclusions are superficial. They do not fully explain how the objectives of the study were achieved. We suggest the authors to approach each part of the stated objective, formulated as we recommend.
Thank you for the suggestion. We have now revised the conclusions accordingly.
We recommend authors to mention the novelty brought by your research compared to the present level of knowledge in the field and underline the originality of their work.
We have now included a line to highlight the novelty of the study as requested.
Reviewer 2 Report
It is useful that the authors have investigated the Recycling-Derived Fertilizers on their P supply for perennial ryegrass (Lolium perenne) in a short term. In general, the MS was written well. The objectives of the study were clearly presented in Introduction. The M &M were described in details. The observed data was statistical analyzed. The conclusions were based on the results. Hence, it is recommended to be published in the present form, except for adding some more explanations on using a shot term test, but not a long term experiment in Discussions
Author Response
It is useful that the authors have investigated the Recycling-Derived Fertilizers on their P supply for perennial ryegrass (Lolium perenne) in a short term. In general, the MS was written well. The objectives of the study were clearly presented in Introduction. The M &M were described in details. The observed data was statistical analyzed. The conclusions were based on the results. Hence, it is recommended to be published in the present form, except for adding some more explanations on using a shot term test, but not a long term experiment in Discussions
Thank you for the positive feedback. We have now included additional lines at the end of the discussion to highlight the need for short- and long-term analyses of struvite and ash applications. Indeed, the current study is the prelude of a long-term study of the same fertilizers in a field trial that is currently in it's last phase.